# RECURRENT CROSS-VIEW OBJECT GEO-LOCALIZATION

## ABSTRACT

Cross-view object geo-localization (CVOGL) aims to determine the location of a specific object in high-resolution satellite imagery given a query image with a point prompt. Existing approaches treat CVOGL as a one-shot detection task, directly regressing object locations from cross-view information aggregation, but they are vulnerable to feature noise and lack mechanisms for error correction. In this paper, we propose **ReCOT**, a **Re**current **C**ross-view **O**bject geo-localization **T**ransformer, which reformulates CVOGL as a recurrent localization task. ReCOT introduces a set of learnable tokens that encode task-specific intent from the query image and prompt embeddings, and iteratively attend to the reference features to refine the predicted location. To enhance this recurrent process, we incorporate two complementary modules: (1) a SAM-based knowledge distillation strategy that transfers segmentation priors from the Segment Anything Model (SAM) to provide clearer semantic guidance without additional inference cost, and (2) a Reference Feature Enhancement Module (RFEM) that introduces a hierarchical attention to emphasize object-relevant regions in the reference features. Extensive experiments on standard CVOGL benchmarks demonstrate that ReCOT achieves state-of-the-art (SOTA) performance while reducing parameters by 60% compared to previous SOTA approaches. Our code will be made available upon acceptance.

## 1 INTRODUCTION

Cross-view object geo-localization (CVOGL) aims to determine the geographic location of a specific object indicated by point prompts in a query image on the reference image Sun et al. (2023). The query images can be captured from devices like phones, autonomous vehicles, robots, and drones, while the reference images are typically high-resolution satellite images. CVOGL is widely used in various applications, such as smart city management Yao et al. (2022), disaster monitoring Chini et al. (2009); Kumar et al. (2013), and robot navigation Singamaneni et al. (2024); Zhai et al. (2024). However, the view gap poses challenges for CVOGL Sun et al. (2023).

Recent cross-view image geo-localization (CVIGL) works Hu et al. (2018); Shi et al. (2019); Zhu et al. (2022a); Yang et al. (2021); Lin et al. (2022) have demonstrated their superiority in handling view gaps. However, CVIGL approaches are fundamentally designed for camera-level localization using retrieval-based approaches Deuser et al. (2023); Zhang et al. (2024b); Shi et al. (2019) or fine-grained approaches Sarlin et al. (2023); Wang et al. (2023). However, CVOGL aims to localize specific objects (e.g., a building with a red roof) captured in the query image, which demands prompt-guided and object-aware prediction. Therefore, in CVOGL scenarios, CVIGL approaches can only provide a nearby location for the indicated object Sun et al. (2023), which is insufficient for precise object-level localization.

To address this, CVOGL approaches emerge recently. Existing approaches Sun et al. (2023); Li et al. (2025); Huang et al. (2025) typically treat CVOGL as a one-shot detection paradigm, where the model directly regresses the object location based on prompt-guided information aggregation, as shown in Fig. 1(a). For example, the recent state-of-the-art (SOTA) approach Huang et al. (2025) aggregates information from cross-view images and prompts to produce a spatial attention matrix, which is used to enhance the reference image features. The enhanced features are then fed into several convolutional layers to regress the object location. While efficient and architecturally simple, such a framework is sensitive to the quality of the enhanced feature Cao et al. (2022). It

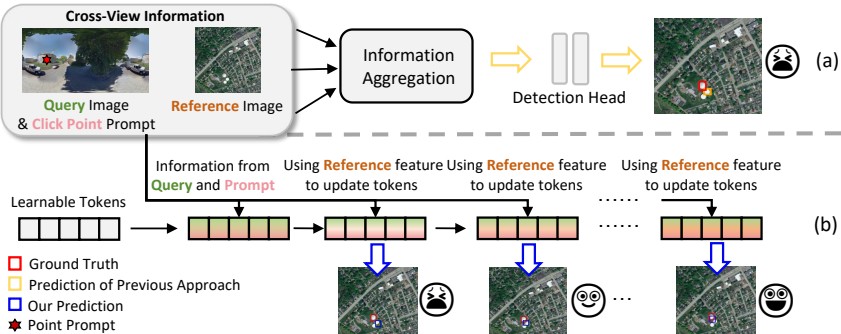

Figure 1: Comparison between the framework of previous CVOGL approaches and ours. (a) Previous approaches treat the CVOGL as a prompt-based detection task, where the model directly regresses the object location based on information aggregation once. (b) Our framework reformulates the CVOGL as a recurrent localization problem, where the model iteratively refines the localization through a set of learnable tokens. Please refer to the zoomed-in view for better visualization.

lacks a correction mechanism for early-stage prediction errors, making them vulnerable to noise in features, $i.e.$, the model cannot give correct localization once the enhanced feature leads to a wrong prediction Cao et al. (2023); Sun et al. (2023).

To cope with this, we propose a **Re**current **C**ross-view **O**bject geo-localization **T**ransformer (ReCOT). Motivated by the success of iterative refinement strategies Yu et al. (2023); Cao et al. (2022; 2023), ReCOT reformulates the CVOGL task as a recurrent localization problem, as shown in Fig. 1(b). This serves as the main difference between previous approaches Sun et al. (2023); Li et al. (2025); Huang et al. (2025) and our framework. Specifically, ReCOT initializes a set of learnable tokens, which interact with the query image feature and prompt embeddings to extract task-specific intent. These tokens then act as recurrent "questioners" that iteratively attend to the enhanced reference image features, progressively extracting object-relevant information and refining the prediction. The proposed recurrent strategy enables our ReCOT to effectively enhance the performance of CVOGL by iteratively refining the initial prediction, as shown in Fig. 1(b).

Nevertheless, the success of this recurrent strategy in our token-driven framework relies on the semantic clarity of the prompts Li et al. (2025) and the quality of reference features Teed & Deng (2020). Specifically, the learnable tokens are first guided by prompt semantics to extract object-relevant intent, then iteratively attend to the reference features to refine the object location in each recurrent step. Therefore, if prompts lack clear semantic intent or reference features are cluttered, the tokens may not accumulate correct task-specific cues across iterations, leading to suboptimal performance. To tackle this, we introduce two complementary methods: (1) a SAM-based knowledge distillation strategy, which injects prior knowledge from large-scale model into the prompt embeddings to boost prompt understanding while avoiding computational cost during inference, and (2) a Reference Feature Enhancement Module (RFEM), which emphasizes object-relevant reference features through hierarchical attention. These components provide clean visual and semantic cues, enabling the tokens to effectively accumulate task-specific information during iterative refinement.

We evaluate ReCOT on the standard CVOGL benchmark Sun et al. (2023). It achieves state-of-the-art (SOTA) performance while reducing parameter count by 60% compared to the previous SOTA approach Huang et al. (2025) (29.9M vs. 74.8M), and runs at a competitive inference speed. In summary, our contributions are as follows:

- We propose ReCOT, a novel framework for CVOGL that reformulates the task as a recurrent localization problem, where learnable tokens iteratively attend to reference features to refine object localization.

- We introduce a SAM-based knowledge distillation strategy that transfers prior knowledge from a large foundation model into the prompt embeddings, providing clearer semantic guidance without adding inference cost.

- We design the RFEM, which leverages a proposed hierarchical attention to highlight object-relevant regions in the reference feature, thereby facilitating the recurrent localization process.

## 2 RELATED WORK

**Cross-View Image Geo-Localization (CVIGL).** CVIGL aims to determine the camera's geographic location by matching a ground-view query image with the most correlated reference image Deuser et al. (2023); Zhang et al. (2024b); Shi et al. (2019) or position Sarlin et al. (2023); Wang et al. (2023); Lentsch et al. (2023). Existing CVIGL approaches can be grouped into metric learning-based methods Lu et al. (2022); Zhu et al. (2022b); Cai et al. (2019); Shi et al. (2020b); Guo et al. (2022); Shi & Li (2022); Shi et al. (2022); Hu et al. (2018); Yang et al. (2021); Lin et al. (2022); Zhu et al. (2022a), which learn viewpoint-invariant features, and geometry-based methods Shi et al. (2020a); Lu et al. (2020); Toker et al. (2021); Liu & Li (2019); Regmi & Shah (2019); Shi et al. (2019), which exploit orientation or structural cues to reduce viewpoint gaps. However, CVIGL methods only provide camera-level localization and cannot accurately pinpoint object-level targets Sun et al. (2023).

**Cross-View Object Geo-Localization (CVOGL).** CVOGL focuses on locating a specific object indicated by prompts in a query image. DetGeo Sun et al. (2023) first formalized this task and proposed a detection-based framework. Subsequent works, such as VaGeo Li et al. (2025) and OCGNet Huang et al. (2025), enhanced cross-view feature aggregation and prompt embedding. Despite progress, existing CVOGL methods still rely on one-shot detection, which is sensitive to noisy features and lacks mechanisms for error correction Cao et al. (2022). In contrast, we reformulate CVOGL as a recurrent localization problem and propose ReCOT to address these limitations.

## 3 METHODOLOGY

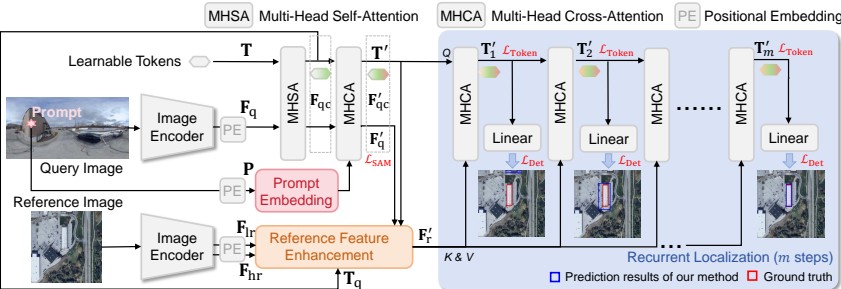

Figure 2: Architecture of our recurrent cross-view object geo-localization transformer (ReCOT). ReCOT reformulates the CVOGL task as a recurrent localization task, which leverages a set of learnable tokens to extract information from cross-view images and prompts to recurrently refine the prediction. Notably, all recurrent steps in ReCOT share the same "MHCA" and "Linear" component.

Fig. 2 presents the architecture of ReCOT, which reformulates CVOGL as a recurrent localization task. A set of learnable tokens is initialized to encode task-specific intent from the query image features and prompt embeddings. These tokens act as recurrent "questioners" that iteratively extract object-relevant cues from the reference features and refine the predicted location through cross-attention mechanisms. Additionally, we introduce two complementary methods to enhance this recurrent process: (1) a SAM-based knowledge distillation strategy, which transfers segmentation priors from the Segment Anything Model (SAM) to enhance prompt semantics, and (2) a Reference Feature Enhancement Module (RFEM), which provides object-relevant reference features through proposed hierarchical attention for the recurrent stage.

### 3.1 RECURRENT LOCALIZATION FRAMEWORK

**Motivation.** As shown in Fig. 1, existing CVOGL approaches follow a one-shot detection paradigm, directly predicting the object location from enhanced reference features. However, such frameworks are often sensitive to feature noise and lack a mechanism for error correction Cao et al. (2022; 2023). Fundamentally, CVOGL can be regarded as a cross-view matching problem, where recurrent strategies have shown superior robustness across domains Cao et al. (2022); Teed & Deng (2020). Inspired by this, we reformulate CVOGL as a recurrent localization process. Moreover, unlike dense

matching tasks Cao et al. (2023); Yu et al. (2023); Edstedt et al. (2024), CVOGL is prompt-driven and focuses on object-level semantic matching. This calls for a representation that can both encode semantic intent and drive iterative refinement. To this end, we draw inspiration from the class token in vision transformers (ViT) Dosovitskiy et al. (2021), and introduce a set of learnable tokens that absorb task-specific semantics from the query and prompt. Acting as semantic carriers, these tokens can recurrently interact with the reference feature to enable step-wise prediction.

**Structure.** We initialize a set of learnable tokens $\mathbf{T} \in \mathbb{R}^{n \times c}$, where $n$ and $c$ denote the number of tokens and the feature dimension, respectively. To enable $\mathbf{T}$ to extract object-relevant information from the reference features, $\mathbf{T}$ needs to first acquire task-specific semantics from the query image feature $\mathbf{F}_q \in \mathbb{R}^{h_q w_q \times c}$ and the point prompt embedding $\mathbf{P} \in \mathbb{R}^c$. Here, $h_q$ and $w_q$ denote the height and width of the query feature, respectively. Specifically, we concatenate $\mathbf{T}$ with $\mathbf{F}_q$ along the spatial dimension, yielding $\mathbf{F}_{qc} \in \mathbb{R}^{(n+h_q w_q) \times c}$. Following standard operations in Vision Transformers (ViT) Dosovitskiy et al. (2021), we apply self-attention to $\mathbf{F}_{qc}$, allowing $\mathbf{T}$ to aggregate global context from the query image. The resulting tokens are denoted as $\mathbf{T}_q$. To further inject object-level intent, the point prompt $\mathbf{P}$ embedded interacts with $\mathbf{F}_{qc}$ through cross-attention. This enables the prompt to semantically guide the tokens, allowing $\mathbf{T}_q$ to further incorporate object-level context. After interaction, we denote the concatenated feature and tokens as $\mathbf{F}'_{qc}$ and $\mathbf{T}'$, respectively. The $\mathbf{F}'_{qc}$ and $\mathbf{T}_q$ are further utilized to enhance reference features in RFEM. The enhanced reference feature are denoted as $\mathbf{F}'_r \in \mathbb{R}^{h_r w_r \times c}$, where $h_r$ and $w_r$ denote the height and width of the reference feature, respectively

After acquiring task-specific semantics from the query and prompt, we use $\mathbf{T}'$ to perform recurrent localization, as shown in Fig. 2. Let $\mathbf{T}'_i$ denote the token state at the $i$-th refinement step, where $i \in [0, 1, 2, \ldots, m]$. We set $m$ to 6 in our work experimentally. At each step, $\mathbf{T}'_i$ attends to the enhanced reference feature $\mathbf{F}'_r$ to extract object-relevant cues and update the task-specific intent. Formally, the update process is defined as

$$\mathbf{T}'_{i+1} = \mathrm{MHCA}(\mathbf{T}'_i, \mathbf{F}'_r) \tag{1}$$

where $\mathrm{MHCA}(\cdot, \cdot)$ denotes the multi-head cross-attention module. Here, $\mathbf{T}'_i$ serves as the query, while $\mathbf{F}'_r$ acts as the key and value. This recurrent attention mechanism enables iterative refinement, where the tokens $\mathbf{T}'$ progressively accumulate task-specific semantics and extract increasingly relevant information from the fixed reference feature $\mathbf{F}'_r$. The $\mathbf{T}'_i$ of each step is fed into a linear layer to predict an updated object location, allowing the model to gradually converge toward a precise localization.

At each refinement step $i$, we introduce a loss $\mathcal{L}_{\mathrm{Token}}$ to guide the generation of $\mathbf{T}'_i$. Specifically, since $\mathbf{T}'_i$ is expected to contain the task-specific intent in cross-view features, it should be able to highlight the required object area on $\mathbf{F}'_r$. Therefore, in each refinement step, we first aggregate $\mathbf{T}'_i$ along the spatial dimension to generate a global embedding $\mathbf{T}''_i$, and use $\mathbf{T}''_i \in \mathbb{R}^{1 \times c}$ and $\mathbf{F}'_r$ to produce an aggregation map $\widehat{\mathbf{m}}_o \in \mathbb{R}^{1 \times h_r \times w_r}$. This can be expressed as

$$\mathbf{T}''_i = \mathrm{Sum}(\mathbf{T}'_i \cdot \mathrm{Softmax}(\mathbf{T}'_i)), \tag{2}$$

$$\widehat{\mathbf{m}}_{oi} = \sigma(\mathbf{T}''_i \mathbf{F}'^{\mathsf{T}}_r), \tag{3}$$

where $\sigma(\cdot)$ and $\mathrm{Softmax}(\cdot)$ denote the sigmoid and softmax function, respectively. $\mathrm{Sum}(\cdot)$ is the summing along the spatial dimension. We then utilize a box-level mask $\mathbf{m}_{oi}$ produced using the ground truth box to supervise the generation of $\widehat{\mathbf{m}}_o$, which can be expressed as

$$\mathcal{L}_{\mathrm{Token}_i}(\mathbf{m}_o, \widehat{\mathbf{m}}_{oi}) = \mathcal{L}_{\mathrm{bce}}(\mathbf{m}_o, \widehat{\mathbf{m}}_{oi}) + \mathcal{L}_{\mathrm{dice}}(\mathbf{m}_o, \widehat{\mathbf{m}}_{oi}), \tag{4}$$

where $\mathcal{L}_{\mathrm{bce}}(\cdot, \cdot)$ and $\mathcal{L}_{\mathrm{dice}}(\cdot, \cdot)$ denote the binary cross-entropy loss and the Dice Loss Milletari et al. (2016), respectively.

**How ReCOT works.** As shown in Fig. 3(a), previous one-shot detection CVOGL approaches Sun et al. (2023); Li et al. (2025); Huang et al. (2025) rely on a single forward information aggregation and are thus sensitive to noisy features Cao et al. (2022; 2023). Our ReCOT adopts a recurrent localization mechanism that iteratively refines predictions. The visualization in Fig. 3(b) of cross-attention weights between tokens and the reference feature reveals the inner dynamics of this refinement process. It can be seen that different tokens focus on different regions of the reference feature, indicating a form of token-level specialization. For object-relevant tokens, their attention gradually

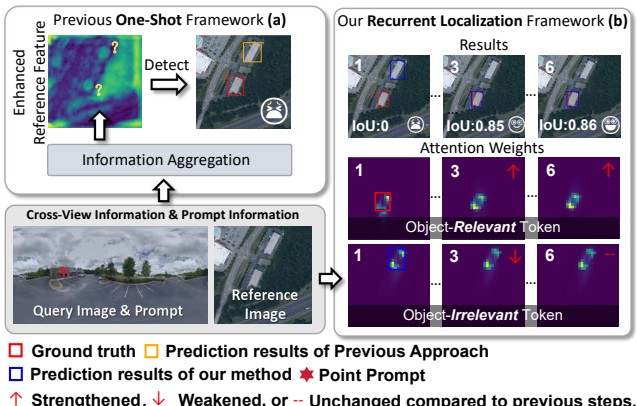

Figure 3: Comparison between the previous one-shot framework and our recurrent localization framework. (a) Previous approaches Li et al. (2025); Sun et al. (2023); Huang et al. (2025) rely on single-shot information aggregation, which is sensitive to noisy enhanced features and often leads to incorrect predictions. (b) Our ReCOT iteratively refines predictions through learnable tokens. The attention weight visualizations show that object-relevant tokens progressively focus and strengthen around the target, while object-irrelevant tokens weaken and stabilize to background patterns. Please refer to the zoomed-in view for better visualization.

concentrates and intensifies around the object region across recurrent steps, reflecting the ability to correct early prediction error and progressively refine the prediction. In contrast, object-irrelevant tokens experience a decrease in their attention responses and eventually stabilize to background patterns once they no longer contribute to the object localization. This behavior highlights the competitive nature of token updates, $i.e.$, multiple tokens initially compete to explain different parts of the reference feature Carion et al. (2020), but those correlated with the object receive positive feedback and their attention weights are amplified over recurrent steps, leading to iterative convergence. Such dynamics demonstrate the effectiveness of recurrent refinement mechanism in suppressing irrelevant regions while enhancing object-relevant cues.

## 3.2 SAM-BASED KNOWLEDGE DISTILLATION

**Motivation.** Point prompt understanding is essential for CVOGL to correctly locate objects. However, point prompt itself suffers from semantic ambiguity, leading to unsatisfactory performance Kirillov et al. (2023). To address this, we propose a SAM-based knowledge distillation strategy to boost the prompt understanding of ReCOT. The incorporation of SAM Kirillov et al. (2023) is motivated by an observation that SAM can give a mask with a clear indication of the required object using point prompts and corresponding images. However, directly applying SAM during inference incurs a large computation overhead. Hence, we leverage predictions of SAM as supervision signals, transferring its knowledge through knowledge distillation.

**Structure.** We extract the query feature $\mathbf{F}'_q$ from the previous concatenated feature $\mathbf{F}'_{qc}$. Notably, the $\mathbf{F}'_q$ has been interacted with prompt embedding and is supposed to contain the object-level semantic. Therefore, we process it using a lightweight convolutional head followed by a sigmoid activation to generate a segmentation map $\widehat{\mathbf{m}}_q$. Meanwhile, we use SAM to generate a pseudo ground-truth mask $\mathbf{m}_{\text{SAM}}$ from the query image and point prompt, This mask is used to supervise the segmentation out of $\mathbf{F}'_q$ via $\mathcal{L}_{\text{SAM}}$, which can be defined as

$$\mathcal{L}_{\text{SAM}}(\mathbf{m}_{\text{SAM}}, \widehat{\mathbf{m}}_q) = \mathcal{L}_{\text{bce}}(\mathbf{m}_{\text{SAM}}, \widehat{\mathbf{m}}_q) + \mathcal{L}_{\text{dice}}(\mathbf{m}_{\text{SAM}}, \widehat{\mathbf{m}}_q), \quad (5)$$

## 3.3 REFERENCE FEATURE ENHANCEMENT MODULE

**Motivation.** In CVOGL, the reference feature $\mathbf{F}_r$ extracted by the backbone encoder is typically generic and background-dominated, lacking object-level specificity before prompt interaction Sun et al. (2023); Li et al. (2025). In our framework, guiding $\mathbf{F}_r$ to focus on the expected object indicated by the prompt can significantly ease the downstream recurrent localization Cao et al. (2023); Teed & Deng (2020). To this end, we propose the Reference Feature Enhancement Module (RFEM),

which enhances reference features into a more object-aware representation $\mathbf{F}'_r$. Unlike previous approaches Sun et al. (2023); Li et al. (2025); Huang et al. (2025) that attempt to clearly highlight the object features through one-shot feature enhancement, RFEM serves as a preparatory module to filter irrelevant features and provide more object-relevant information for subsequent recurrent localization. The key of RFEM lies in its hierarchical attention design. We first perform spatial attention to highlight the query-relevant regions in the reference feature, as the expected object is likely confined to these regions. We then apply cross-attention guided by the query and prompt cues to refine these regions with object-level semantics. This spatial-to-cross hierarchy yields a cleaner and more informative reference feature $\mathbf{F}'_r$, which significantly benefits the iterative localization process of ReCOT.

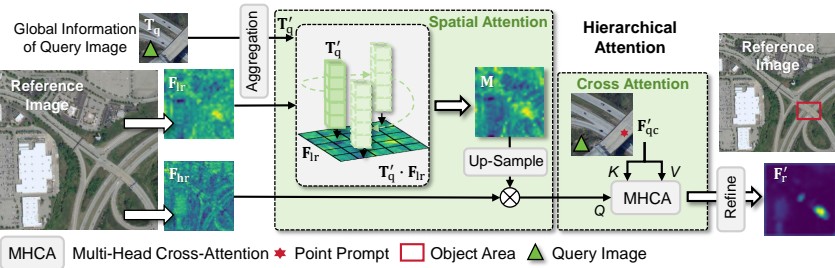

Figure 4: Architecture of reference feature enhancement module (RFEM). RFEM enhances reference features through a hierarchical attention pipeline to obtain $\mathbf{F}'_r$.

**Structure.** Specifically, as illustrated in Fig. 4, we utilize two levels of reference features, *i.e.*. a low-resolution semantic feature $\mathbf{F}_{lr} \in \mathbb{R}^{h_r \times w_r \times c}$ that captures scene semantics, and a high-resolution detailed feature $\mathbf{F}_{hr} \in \mathbb{R}^{2h_r \times 2w_r \times c}$ that preserves fine-grained spatial structures Zhang et al. (2024c).

*Spatial Attention:* We leverage the spatial attention to highlight query-relevant features. Specifically, We first aggregate the query tokens $\mathbf{T}_q$ into a global descriptor $\mathbf{T}'_q$ (Eq. (2)). Notably, the $\mathbf{T}'_q$ contains only the global information of the query image without prompt guidance. This descriptor correlates with $\mathbf{F}_{lr}$ via a dot-product operation to produce a spatial attention map as

$$\mathbf{M} = \sigma(\mathbf{T}'_q \mathbf{F}_{lr}^\mathsf{T}), \tag{6}$$

which highlights query-relevant regions. The attention map is then up-sampled and applied to $\mathbf{F}_{hr}$ by element-wise multiplication, suppressing irrelevant background and narrowing the focus to potential object areas.

*Cross Attention:* We leverage the cross-attention to incorporate detailed query features and prompt semantics for further refining the reference feature. The filtered $\mathbf{F}_{hr}$ is subsequently refined via cross-attention with $\mathbf{F}_{qc}$, which encodes fine-grained prompt cues and detailed query features. This cross-attention step sharpens the object-relevant responses and injects object-level semantics into the reference representation.

Finally, the updated feature is down-sampled and refined through spatial attention Woo et al. (2018) and self-attention Dosovitskiy et al. (2021) to generate the $\mathbf{F}'_r$, which is then passed to the recurrent localization stage in ReCOT.

**Why multi-resolution reference features?** The reference image typically contains both global scene information and fine-grained object details. The low-resolution semantic reference feature $\mathbf{F}_{lr}$ extracted from deeper layers of the backbone encodes more scene-level context but lose spatial details due to down-sampling. Therefore, we utilize $\mathbf{F}_{lr}$ to enhance query-relevant regions. Conversely, the high-resolution reference features $\mathbf{F}_{hr}$ preserve fine structural details but are dominated by background noise. Thus, it needs to be refined by the attention map produced by $\mathbf{F}_{lr}$, and then RFEM can utilize it to perform object-level enhancement. This design leverages advantages of multi-resolution features, which is crucial for object-level feature enhancement in CVOGL Huang et al. (2025).

## 3.4 LOSS FUNCTION

Our training objective $\mathcal{L}$ is defined as a weighted sum of the localization loss $\mathcal{L}_{local}$ and the auxiliary loss $\mathcal{L}_{aux}$. It can be defined as

$$\mathcal{L} = \mathcal{L}_{local} + \alpha \mathcal{L}_{aux}, \tag{7}$$

where $\mathcal{L}_{local}$ denotes the sum of DETR-style detection losses $\mathcal{L}_{Det}$ computed at each recurrent localization step (see Fig. 2). The auxiliary loss $\mathcal{L}_{aux}$ is the sum of $\mathcal{L}_{Token_i}$ across all recurrent steps and the SAM-based distillation loss $\mathcal{L}_{SAM}$. The balancing coefficient $\alpha$ is set to 1 in all experiments.

# 4 EXPERIMENT

## 4.1 DATASETS

**CVOGL-DetGeo** dataset Sun et al. (2023) divides the task into "Ground $\rightarrow$ Satellite" task and "Drone $\rightarrow$ Satellite" task. This dataset is the only public dataset for the relatively new task CVOGL. It contains 6,239 pairs of "Ground $\rightarrow$ Satellite" view and 6,239 pairs of "Drone $\rightarrow$ Satellite" view query and reference images. The ground view query images, drone view query images, and satellite view reference images are sized to $512 \times 256$, $256 \times 256$, and $1024 \times 1024$, respectively. Each cross-view task uses 4,343 pairs for training, 923 pairs for validation, and 973 pairs for testing. Our experiments utilize the training set to train the model and select the best model on the validation set for evaluation on the test set.

## 4.2 EVALUATION METRICS

Following the pioneering work DetGeo Sun et al. (2023), we adopt Acc@0.25 and Acc@0.50 for evaluation. For each pair of the query and reference image, we select the box with the highest confidence output by our model as the final prediction box. Additionally, we use the parameter and frames per second (FPS) to show the model efficiency. Higher Acc@0.25, higher Acc@0.50, Higher FPS, and fewer parameter denote better performance. Please refer to the appendix for more detailed introduction of evaluation metrics.

## 4.3 IMPLEMENTATION DETAILS

Please refer to the appendix for detailed introduction of implementation details.

## 4.4 ABLATION STUDY

**Effect of Total Recurrent Localization Steps.** The role of the recurrent localization lies in refining and correcting early predictions. Table 1 investigates the impact of varying the total number of recurrent steps $m$ on ReCOT performance. For the Ground→Satellite task, increasing $m$ from 1 (one-shot prediction) to 5 yields consistent improvements, with the best performance achieved at $m = 5$. In contrast, for the Drone→Satellite task, the Drone and Satellite images share similar top-down geometry and exhibit cleaner alignment, a single forward pass already produces a strong estimate. Its performance saturates earlier, with $m = 3$ giving the relatively optimal results. This discrepancy indicates that the optimal number of recurrent steps is task-dependent due to differences in viewpoint variations and feature alignment difficulty across scenarios. Additionally, further increasing $m$ beyond the optimal point does not bring additional gains and may slightly degrade performance, which can be attributed to over-refinement and overfitting in deeper iterations Hur & Roth (2019); Cao et al. (2023); Yu et al. (2023). Based on the results, we set $m = 5$ for the Ground→Satellite task, while $m = 3$ for the Drone→Satellite task, and keep this setting in other experiments.

**Effect of Components in ReCOT.** Table 2 presents the ablation study of the key components in ReCOT on the CVOGL-DetGeo test set. Removing the RFEM module (w/o RFEM) leads to a noticeable performance drop, confirming that early reference feature enhancement is critical for guiding tokens to focus on relevant regions. Similarly, removing the SAM-based distillation loss $\mathcal{L}_{SAM}$ (w/o $\mathcal{L}_{SAM}$) results in performance degradation, indicating the importance of accurate prompt semantics understanding. The full ReCOT model achieves the best performance across both scenarios. In addition, removing the token-guidance loss $\mathcal{L}_{Token}$ (w/o $\mathcal{L}_{Token}$) also degrades performance,

which highlights its role in encouraging the learnable tokens to accurately capture object-relevant areas during recurrent refinement. These results collectively validate that RFEM, $\mathcal{L}_{\text{SAM}}$, and $\mathcal{L}_{\text{Token}}$ complement each other, contributing to the robust performance of ReCOT.

Table 1: Ablation study on the recurrent localization steps $m$ of ReCOT in terms of Acc@0.25(%) ↑ and Acc@0.50(%) ↑ on the test set of CVOGL-DetGeo dataset. Bold and Underline indicate the best and second-best results, respectively.

| Steps $m$ | Ground→Satellite | | | Drone→Satellite | | |
|---|---|---|---|---|---|---|
| | Acc@0.25 | Acc@0.50 | FPS | Acc@0.25 | Acc@0.50 | FPS |
| 1 | 49.74 | 46.25 | 26.9 | **78.31** | 71.74 | 27.2 |
| 2 | 51.08 | 47.17 | 26.2 | 78.21 | 72.15 | 26.5 |
| 3 | 51.28 | 47.58 | 25.6 | 78.21 | 72.35 | 25.7 |
| 4 | 51.39 | 47.89 | 24.8 | 77.90 | **72.56** | 24.9 |
| 5 | **52.00** | **48.10** | 24.1 | 77.60 | 72.05 | 24.3 |
| 6 | 51.70 | **48.10** | 23.4 | 77.49 | 71.84 | 23.5 |

Table 2: Ablation study on the components of ReCOT in terms of Acc@0.25(%) ↑, Acc@0.50(%) ↑, and FPS ↑ on the test set of CVOGL-DetGeo dataset.

| Component | Ground→Satellite | | Drone→Satellite | |
|---|---|---|---|---|
| | Acc@0.25 | Acc@0.50 | Acc@0.25 | Acc@0.50 |
| w/o RFEM | 46.45 | 42.24 | 50.15 | 46.04 |
| w/o $\mathcal{L}_{\text{SAM}}$ | 49.74 | 44.81 | 70.91 | 65.78 |
| w/o $\mathcal{L}_{\text{Token}}$ | 50.57 | 47.58 | 72.05 | 66.80 |
| ReCOT (Ours) | 52.00 | 48.10 | 78.21 | 72.35 |

**Effect of Components Within the RFEM.** Table 3 investigates the contributions of components within the RFEM. Removing the weight matrix $\mathbf{M}$ (w/o $\mathbf{M}$) leads to a noticeable performance drop, particularly in the Drone→Satellite setting. This confirms that the spatial attention stage benefit the reference feature enhancement. Furthermore, replacing the low-resolution semantic feature $\mathbf{F}_{\text{lr}}$ with the high-resolution feature $\mathbf{F}_{\text{hr}}$ and replacing $\mathbf{F}_{\text{hr}}$ with $\mathbf{F}_{\text{lr}}$ both result in performance degradation, highlighting the importance of leveraging multi-resolution reference features in RFEM. Please refer to the appendix for more ablation study results.

Table 3: Ablation study inside the RFEM in terms of Acc@0.25(%) ↑ and Acc@0.50(%) ↑ on the test set of CVOGL-DetGeo dataset.

| RFEM | Ground→Satellite | | Drone→Satellite | |
|---|---|---|---|---|
| | Acc@0.25 | Acc@0.50 | Acc@0.25 | Acc@0.50 |
| w/o $\mathbf{M}$ | 51.18 | 47.48 | 75.44 | 70.30 |
| Replace $\mathbf{F}_{\text{lr}}$ with $\mathbf{F}_{\text{hr}}$ | 48.00 | 44.19 | 75.85 | 70.09 |
| Replace $\mathbf{F}_{\text{hr}}$ with $\mathbf{F}_{\text{lr}}$ | 47.49 | 43.28 | 75.64 | 67.11 |
| ReCOT (Ours) | 52.00 | 48.10 | 78.21 | 72.35 |

## 4.5 Comparison Results

**Quantitative Results.** Table 4 compares ReCOT with existing SOTA cross-view localization approaches on the CVOGL-DetGeo dataset. ReCOT consistently outperforms all competitors in both the Ground→Satellite and Drone→Satellite settings, achieving new SOTA performance across almost all metrics in the test set. Despite relatively lower performance on the validation set for Ground→Satellite, it maintains the best performance on the test set, indicating stronger generalization ability. Notably, as shown in Table 4, these performance gains are achieved with significantly fewer parameters, representing a 60% reduction in model size. ReCOT is a little slower in inference speed compared to the previous CVOGL works Sun et al. (2023); Huang et al. (2025) due to iterative framework Cao et al. (2023). However, it still achieves a competitive inference speed, making ReCOT both efficient and scalable for real-world applications. Compared to the Drone→Satellite setting, the improvement of ReCOT on Ground→Satellite is relatively smaller. This is mainly due to the larger viewpoint gap and severe occlusions in ground-view images Sun et al. (2023), where

objects are often partially visible or obstructed by surrounding structures. Moreover, ground images typically contain more background clutter, making recurrent localization harder. We believe integrating geometric priors (*e.g.*, camera pose estimation) or multi-view fusion could further boost Ground→Satellite performance. Please refer to the appendix for more comparison results.

Table 4: Comparisons in terms of Acc@0.25(%) ↑, Acc@0.50(%) ↑, Parameter (M) ↓, and FPS ↑ on the CVOGL-DetGeo dataset. Following Table 1, we set $m = 5$ and $m = 3$ for the Ground→Satellite and Drone→Satellite task, respectively. The inference speed is test on the Drone→Satellite ($m = 3$) task using one NVIDIA GeForce RTX 4090 GPU. Bold and Underline indicate the best and second-best results, respectively.

| Method | Ground→Satellite | | | | Drone→Satellite | | | | Param | FPS |
|---|---|---|---|---|---|---|---|---|---|---|
| | Test Set | | Validation Set | | Test Set | | Validation Set | | | |
| | Acc@0.25 | Acc@0.50 | Acc@0.25 | Acc@0.50 | Acc@0.25 | Acc@0.50 | Acc@0.25 | Acc@0.50 | | |
| CVM-Net Hu et al. (2018) | 4.73 | 0.51 | 5.09 | 0.87 | 20.14 | 3.29 | 20.04 | 3.47 | – | – |
| RK-Net Lin et al. (2022) | 7.40 | 0.82 | 8.67 | 0.98 | 19.22 | 2.67 | 19.94 | 3.03 | – | – |
| L2LTR Yang et al. (2021) | 10.69 | 2.16 | 12.24 | 1.84 | 38.95 | 6.27 | 38.68 | 5.96 | – | – |
| Polar-SAFA Shi et al. (2019) | 20.66 | 3.19 | 19.18 | 2.71 | 37.41 | 6.58 | 36.19 | 6.39 | – | – |
| TransGeo Zhu et al. (2022a) | 21.17 | 2.88 | 21.67 | 3.25 | 35.05 | 6.47 | 34.78 | 5.42 | – | – |
| SAFA Shi et al. (2019) | 22.20 | 3.08 | 20.59 | 3.25 | 37.41 | 6.58 | 36.19 | 6.39 | – | – |
| GeoDTR+ Zhang et al. (2024b) | 14.19 | 5.14 | 14.08 | 1.95 | 16.03 | 4.73 | 15.71 | 3.68 | – | – |
| DetGeo Sun et al. (2023) | 45.43 | 42.24 | 46.70 | 43.99 | 61.97 | 57.66 | 59.81 | 55.15 | 73.8 | **29.5** |
| VaGeo Li et al. (2025) | 48.21 | 45.22 | 47.56 | **44.42** | 66.19 | 61.87 | 64.25 | 59.59 | – | – |
| OCGNet Huang et al. (2025) | 51.49 | 47.69 | **48.54** | 44.20 | 68.39 | 63.93 | 66.52 | 61.86 | 74.8 | 27.7 |
| ReCOT (Ours) | **52.00** | **48.10** | 48.43 | 43.66 | **78.21** | **72.35** | **74.00** | **67.17** | 29.9 | 25.7 |

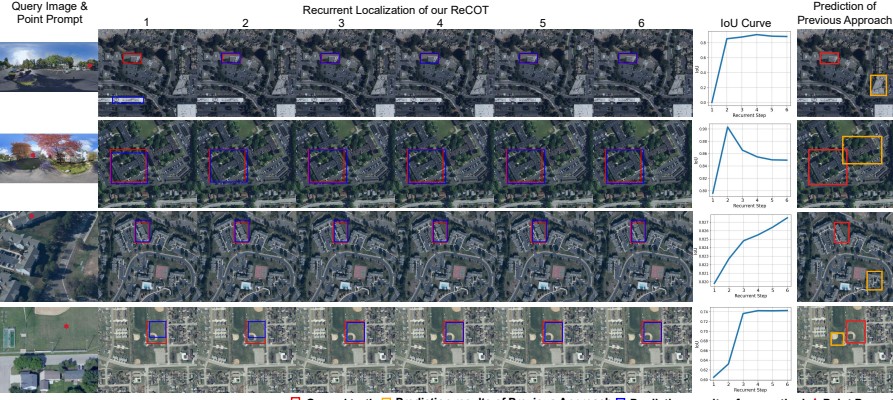

Figure 5: Visualization of some representative results produced by our ReCOT and the previous work Sun et al. (2023). Please refer to the zoomed-in view for better visualization.

**Qualitative Results.** Fig. 5 visualizes some representative results. As the recurrent steps proceed, our model progressively refines the bounding boxes, leading to higher-quality localization compared to the single-shot prediction of the previous approach Sun et al. (2023). However, the optimal number of recurrent steps varies across different scenarios, and excessive iterations may cause over-refinement Hur & Roth (2019), resulting in a slight performance drop. Therefore, based on the ablation results in Table 1, we set the number of recurrent steps to 5 for Ground→Satellite and 4 for Drone→Satellite, which provides the best trade-off between accuracy and stability.

## 5 CONCLUSION

In this paper, we propose ReCOT, which reformulates the CVOGL task as a recurrent localization problem. ReCOT introduces learnable tokens to encode task-specific semantics and recurrently refine predictions. We further incorporate a SAM-based knowledge distillation scheme to improve prompt understanding without incurring additional inference costs, and a RFEM to produce object-aware reference features via a hierarchical attention strategy. Extensive experiments on the CVOGL-DetGeo benchmark demonstrated that ReCOT achieves state-of-the-art performance with significantly fewer parameters and competitive inference speed.

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

## A APPENDIX

### A.1 MORE EXPERIMENTAL RESULTS

#### A.1.1 DETAILED INTRODUCTION OF EVALUATION METRICS.

**Definition of Acc@0.25 and Acc@0.50.** The Acc@0.25 and Acc@0.50 measure the prediction accuracy under a specific intersection over union (IoU) threshold $t$ between the predicted box $b_p$ and ground box $b_g$ as

$$\text{Acc@}t = \frac{1}{n} \sum_{i=1}^{n} \psi(t), \tag{8}$$

where

$$\psi(t) = \left\{ \begin{array}{l} 1, \text{IoU}(b_p, b_g) \geq t \\ 0, \text{otherwise} \end{array} \right., \tag{9}$$

$$\text{IoU}(b_p, b_g) = \frac{|b_p \cap b_g|}{|b_p \cup b_g|}. \tag{10}$$

#### A.1.2 IMPLEMENTATION DETAILS

We conduct all experiments on four NVIDIA GeForce RTX 4090 GPUs, with implementations based on PyTorch Paszke et al. (2019). For training, we adopt the AdamW Loshchilov & Hutter (2017) as the optimizer and set the initial learning rate to 0.0025, the weight decay rate to 0.0001, and the batch size to 16. We train our network for 300 epochs for all the experiments. Since CVOGL is a relatively new task, we follow the pioneering work Sun et al. (2023) and select five CVIGL approaches Hu et al. (2018); Lin et al. (2022); Yang et al. (2021); Shi et al. (2019); Zhu et al. (2022a) and the existing CVOGL approaches Sun et al. (2023); Li et al. (2025); Huang et al. (2025) as our comparison methods. The results of CVIGL comparison methods can be found in previous works Sun et al. (2023); Huang et al. (2025); Li et al. (2025). Additionally, we adopt swin transformer (Swin-t) Liu et al. (2021) as the image encoder for its superior performance on various fields Zhang et al. (2024a); Shi et al. (2024); Chi et al. (2023); Zamir et al. (2022). We set the hyper-parameter $m$ to 6 during training.

#### A.1.3 MORE ABLATION STUDY RESULTS

**Detailed explanation of performance degradation with excessive steps in our recurrent localization.** The recurrent mechanism in ReCOT aims to progressively refine the predicted location. However, as reported in Table 1, excessive recurrent steps may sightly degrade performance. This is because our recurrent localization can be interpreted as predicting and applying residual corrections to the bounding box. After the prediction becomes sufficiently accurate, the remaining residuals are mainly high-frequency noise or background fluctuations. As observed in other iterative refinement works Cao et al. (2022; 2023), continuing to update on such residuals may amplify noise and introduce attention drift, causing the learnable tokens to overfit spurious cues rather than consolidate stable object semantics Cai & Vasconcelos (2021). From an optimization perspective, when the objective is already close to its minimum, further iterations may overshoot or oscillate around the optimum Teed & Deng (2020), leading to a slight decline in IoU.

**How to determine $m$ on new datasets?** To determine $m$ on a new dataset, one can monitor IoU gain per step on a validation set and stop when the gain becomes negligible or unstable. Tasks with large viewpoint gaps typically require larger $m$, while nearly aligned views (*e.g.*, Drone$\rightarrow$Satellite) benefit from smaller $m$ for a better balance of accuracy and efficiency.

**Effect of the parameter $\alpha$.** Table 5 shows how the performance of ReCOT varies with different values of the balancing coefficient $\alpha$, which controls the weight between the localization loss $\mathcal{L}_{\text{local}}$ and the auxiliary loss $\mathcal{L}_{\text{aux}}$. We observe that $\alpha = 1$ yields the best performance on both Ground$\rightarrow$Satellite and Drone$\rightarrow$Satellite. A smaller value ($\alpha = 0.1$) weakens the supervision of token alignment and SAM distillation, slightly reducing accuracy. Conversely, a larger value ($\alpha = 10$) overemphasizes auxiliary objectives, causing a notable drop. These results indicate that $\alpha = 1$ achieves the optimal trade-off.

Table 5: Ablation study on the hyper-parameter $\alpha$ in terms of Acc@0.25(%) $\uparrow$ and Acc@0.50(%) $\uparrow$ on the test set of CVOGL-DetGeo dataset.

| $\alpha$ | Ground$\rightarrow$Satellite | | Drone$\rightarrow$Satellite | |
|---|---|---|---|---|
| | Acc@0.25 | Acc@0.50 | Acc@0.25 | Acc@0.50 |
| 0.1 | 51.28 | 48.10 | 75.13 | 69.68 |
| 1 | 52.00 | 48.10 | 78.21 | 72.35 |
| 10 | 48.51 | 45.02 | 76.05 | 70.20 |

**Effect of the hyper-parameter $n$.** Table 6 investigates the impact of the token number $n$ on the performance of ReCOT. We observe a clear performance gain when increasing $n$ from 1 to 100. This demonstrates that a sufficient number of tokens provides a richer representation of task-specific intent and better coverage of cross-view semantics, which benefits the recurrent refinement process. However, when $n$ is further increased to 200, performance drops across all metrics. This decline is likely due to over-parameterization and token redundancy, which can introduce noise and hinder effective attention learning Dosovitskiy et al. (2021), as also observed in token-scaling studies Dosovitskiy et al. (2021); Touvron et al. (2021). Therefore, we set $n$ to 100 in this work.

Table 6: Ablation study on the hyper-parameter $n$ in terms of Acc@0.25(%) $\uparrow$ and Acc@0.50(%) $\uparrow$ on the test set of CVOGL-DetGeo dataset.

| $n$ | Ground$\rightarrow$Satellite | | Drone$\rightarrow$Satellite | |
|---|---|---|---|---|
| | Acc@0.25 | Acc@0.50 | Acc@0.25 | Acc@0.50 |
| 1 | 47.89 | 43.37 | 74.31 | 68.86 |
| 100 | 52.00 | 48.10 | 78.21 | 72.35 |
| 200 | 50.46 | 47.28 | 72.25 | 67.01 |

Table 7: Comparisons between DetGeo Sun et al. (2023), DetGeo using Swin-t Liu et al. (2021) (DetGeo[*]), and our methods in terms of Acc@0.25(%) $\uparrow$ and Acc@0.50(%) $\uparrow$ on the test set of CVOGL-DetGeo dataset.

| Method | Ground$\rightarrow$Satellite | | Drone$\rightarrow$Satellite | |
|---|---|---|---|---|
| | Acc@0.25 | Acc@0.50 | Acc@0.25 | Acc@0.50 |
| DetGeo | 45.43 | 42.24 | 61.97 | 57.66 |
| DetGeo[*] | 46.01 | 41.44 | 66.60 | 56.11 |
| VaGeo | 48.21 | 45.22 | 66.19 | 61.87 |
| OCGNet | 51.49 | 47.69 | 68.39 | 63.93 |
| ReCOT (Ours) | **52.00** | **48.10** | **78.21** | **72.35** |

### A.1.4 MORE COMPARISON RESULTS.

**Quantitative Results.**

As shown in Table 7, we replace the backbone of DetGeo Sun et al. (2023) with Swin-t Liu et al. (2021) for a more comprehensive and fair comparison. While the upgraded DetGeo* shows slight improvements on the Acc@0.25, it still lags behind our ReCOT by a large margin across all evaluation metrics. Moreover, the backbone replacement does not lead to consistent gains, as DetGeo* fails to outperform other recent CVOGL methods Li et al. (2025); Huang et al. (2025) in Table 7. This demonstrates that the key factor of CVOGL does not lie in the backbone. In contrast, our proposed ReCOT achieves superior performance through a more effective and task-aligned framework, highlighting the importance of architectural innovations for CVOGL.

**Qualitative Results.**

Fig. 6 provides more visualization results of the cross attention weights beteen the token and reference features during recurrent process. As the recurrent steps progress, object-relevant tokens gradually focus and strengthen around the expected object, enabling the bounding box to converge toward the correct location. In contrast, object-irrelevant tokens weaken and gradually stabilize over iterations. This behavior highlights the ability of ReCOT to dynamically disentangle object-focused information from irrelevant features, which is a key factor driving the success of our recurrent localization strategy.

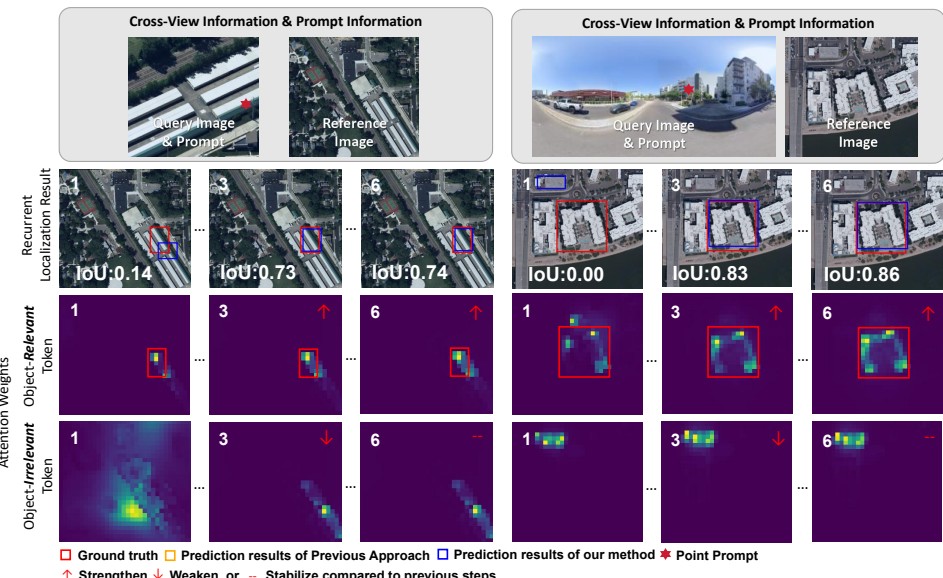

Figure 6: Visualizations of how our ReCOT works. The object-relevant tokens progressively focus and strengthen around the indicated object, while object-irrelevant tokens weaken and stabilize to background patterns.

## A.2 LIMITATIONS AND FUTURE WORK

As shown in Fig. 7, some failure cases of ReCOT are caused by ambiguous or imprecise point prompts, which often highlight only a part of the object rather than its entirety. This ambiguity misguides the token-driven recurrent refinement process, leading to suboptimal localization results. In the future, we will incorporate multi-modal prompts (*e.g.*, textual descriptions or bounding boxes) to provide richer and more accurate prompt semantics. Such multi-modal guidance could reduce ambiguity and further improve the robustness and precision of recurrent localization.

## A.3 STATEMENT

### A.3.1 ETHICS STATEMENT

This work addresses cross-view object geo-localization in the recurrent manner. All experiments are conducted on publicly available datasets that do not contain sensitive information. The authors affirm that this work complies with the code of ethics.

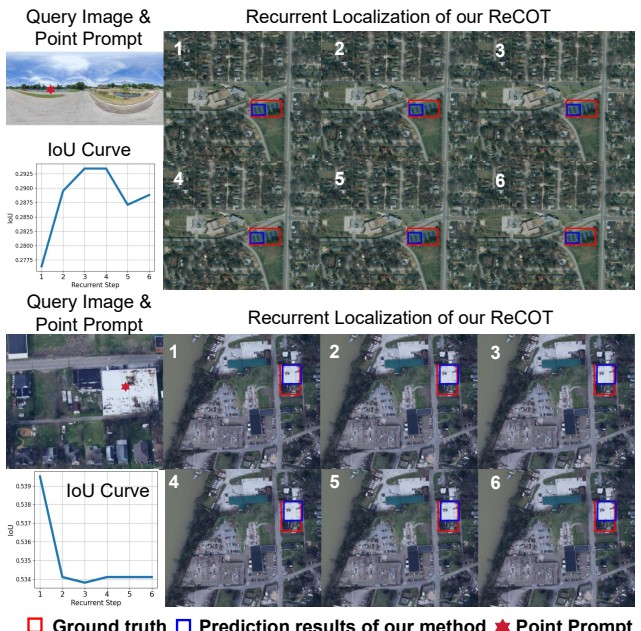

Figure 7: Examples of failure cases. Please refer to the zoomed-in view for better visualization.

### A.3.2 REPRODUCIBILITY STATEMENT

We provide the source code in the supplementary material. Upon acceptance, we will release the source code, repurposed datasets, and model weights to ensure the reproducibility of the experimental results.

### A.3.3 THE USE OF LARGE LANGUAGE MODEL (LLM)

We used the LLM (*e.g.*, ChatGPT) only to aid English writing and polish the presentation of this paper (grammar, clarity, and minor wording suggestions). The research ideas, algorithms, theoretical analysis, experiment design, implementation, and all reported results were entirely developed and validated by the authors without automated assistance. No content was directly copied from model outputs, and all technical claims have been carefully verified by the authors.

