# OpenReview forum: "Recurrent Cross-View Object Geo-Localization"
_ICLR.cc/2026/Conference — ICLR 2026 Conference Withdrawn Submission_

### Official Review · Reviewer_5hQx · 2025-10-25

**Soundness:** 3
**Presentation:** 3
**Contribution:** 2
**Rating:** 4
**Confidence:** 3

**Summary:**

This paper aims to tackle the cross-view object geo-localization task with an iterative localization pipeline called ReCOT. ReCOT introduces a set of learnable tokens that encode task-specific intent from the query image and prompt embeddings, and iteratively attend to the reference features to refine the predicted location. Two complementary modules are then proposed to enhance the performance: (1) a SAM-based knowledge distillation strategy for fine-grained supervision of RoIs, (2) a Reference Feature Enhancement Module (RFEM) to enhance the object-relevant features in the feature map of reference images. Experiments on a single benchmark show improved accuracy and reduced model size compared with state-of-the-art methods.

**Strengths:**

- This paper proposes a new pipeline for cross-view object geo-localization (CVOGL), which iteratively updates the predicted object location in the reference image by distilling the semantics of the SAM model, and performing multi-scale ref-query cross-attention. This proposed method offers a novel and intuitive solution to the CVOGL task.
- The presented CVOGL pipeline is clearly presented and easy to follow.
- Experiments on the CVOGL-DetGeo benchmark demonstrate that the proposed method achieves state-of-the-art performance with significantly fewer parameters and competitive inference speed.

**Weaknesses:**

- Although it is mentioned in the introduction that CVOGL can be “widely used in various applications”, the real-world significance and concrete application scenarios of this task need further clarification. As written, the task feels more assumed than demand-driven.
- As a relatively niche field, it may be helpful to have a clear definition of the task to help readers better understand the input, output, and objective of this task before introducing the method. Since the approach is claimed as a reformulation, explicitly stating how the formulation changes would also help readers better follow the core contribution.
- It is interesting to leverage the SAM to provide fine-grained mask supervision for objects during the iterative optimization process. However, since SAM is mainly optimized for natural images, it is not clear how accurate the generated mask is given panorama or drone image inputs. Moreover, the introduction of the SAM and distillation strategy would likely increase the computation at the training stage, while related discussions are missing in the paper.
- Table 4 adapts image geo-localization methods to object geo-localization, but these methods were not originally designed to localize objects in reference images. How they were adapted (training targets, proposal generation, matching strategy) could significantly affect results, while key implementation details are missing.
- The discussion on the advantage of the proposed method over previous methods is somewhat vague in related works.
- Results are reported on a single and relatively small dataset, which limits the convincingness of the experimental demonstration.
- Formatting issues:
The manuscript appears not to use the guideline-recommended \citep{}.
Several references are missing publication venues.

**Questions:**

I am open to discussion and willing to change my opinion if these weaknesses can be properly addressed

---

### Official Review · Reviewer_bZgt · 2025-10-26

**Soundness:** 3
**Presentation:** 4
**Contribution:** 3
**Rating:** 6
**Confidence:** 3

**Summary:**

This paper addresses the cross-view object geo-localization problem by reformulating it as a recurrent localization process rather than a one-shot prediction. The proposed ReCOT model iteratively refines object positions via cross-attention, incorporates SAM-based knowledge distillation for semantic enhancement, and introduces a reference feature enhancement module for multi-scale fusion. Experiments on the DetGeo dataset demonstrate superior accuracy and parameter efficiency compared to prior methods, validating the effectiveness of recurrent reasoning in cross-view localization.

**Strengths:**

+ Novel problem formulation. Reformulating cross-view object geo-localization as a recurrent refinement task is a fresh perspective that effectively addresses the limitations of one-shot prediction approaches.

+ Well-structured and interpretable architecture. The recurrent token mechanism enables iterative attention focusing and error correction, while the Reference Feature Enhancement Module (RFEM) provides robust multi-scale feature fusion.

+ Efficient use of foundation model knowledge. The SAM-based distillation strategy cleverly transfers segmentation priors into the model without adding inference cost.

**Weaknesses:**

+ Inconsistent setting of recurrent steps (m) between training and inference. The paper mentions different numbers of recurrent iterations (e.g., m=6 for training but m=3 or 5 for inference) without clearly explaining the rationale or impact.  It remains unclear whether the model was trained with shared weights across iterations or if the inference depth differs intentionally.

+ Heavy reliance on SAM-based distillation. The success of the model largely depends on the quality of SAM-generated pseudo-masks. No analysis is provided on how errors or inconsistencies in SAM segmentation affect downstream localization accuracy. Moreover, there is no comparison with alternative distillation sources.

**Questions:**

+ Parameter sharing and expressive limitation. You mention that all recurrent steps share the same parameters. Could you clarify how this design affects model expressivity? Did you experiment with an unshared-parameter variant (i.e., multiple distinct Transformer layers) to test whether parameter sharing sacrifices representational capacity?
+ Stability and convergence of the recurrent process. How stable is the recurrent update process across iterations? Does the performance consistently improve over steps, or does it oscillate or saturate?
+ SAM distillation robustness. Since your method relies on SAM-generated pseudo-masks for distillation, how robust is the model when the masks are inaccurate (e.g., occlusion, cluttered background, or small objects)? Have you evaluated how the localization performance degrades as the SAM mask quality decreases?

---

### Official Review · Reviewer_a4Ve · 2025-10-27

**Soundness:** 2
**Presentation:** 3
**Contribution:** 2
**Rating:** 4
**Confidence:** 3

**Summary:**

This paper tackles Cross-View Object Geo-Localization (CVOGL), locating a specific object (e.g., a building or car) in high-resolution satellite imagery given a ground or drone query image and a point prompt.
The authors propose ReCOT, a Recurrent Cross-view Object geo-localization Transformer, which reformulates the task as an iterative refinement process rather than a one-shot regression.

**Strengths:**

- Reformulating CVOGL as a recurrent problem is conceptually elegant and directly addresses error-correction and noise sensitivity in one-shot methods.
- Strong empirical performance. ReCOT outperforms all prior works (DetGeo, VaGeo, OCGNet) on both Ground→Satellite and Drone→Satellite tasks.

**Weaknesses:**

- Limited benchmark diversity: The experimental evaluation is conducted exclusively on the CVOGL-DetGeo dataset. It would strengthen the paper to include results on additional benchmarks (e.g., CVUSA, CVACT, or other city-scale datasets) to demonstrate the generalization ability of ReCOT across diverse domains and scene types.
- Underexplained SAM distillation process: The details of generating pseudo masks from SAM and aligning them with prompt embeddings are insufficiently described. Please specify the SAM version, hyperparameter settings (e.g., mask threshold, prompt type), and any post-processing steps used. Additionally, the paper should clarify how the quality of the pseudo masks and the distilled segmentation maps was assessed or verified during training.
- Incomplete efficiency analysis: In Table 1, the computational efficiency under different recurrent localization steps m should be reported. Presenting FLOPs, latency, or inference time for varying m values would help readers understand the trade-off between localization accuracy and computational cost.

**Questions:**

Please refer to the weakness above.

---

### Official Review · Reviewer_S6zQ · 2025-11-01

**Soundness:** 2
**Presentation:** 3
**Contribution:** 2
**Rating:** 2
**Confidence:** 5

**Summary:**

This paper addresses the task of Cross-View Object Geo-Localization (CVOGL). Its primary novel contribution is the reformulation of this task from a standard one-shot detection paradigm into a recurrent localization problem. The proposed method, ReCOT, uses learnable tokens as "recurrent questioners" to iteratively refine a location prediction. This new framework is shown to achieve state-of-the-art (SOTA) performance.

**Strengths:**

1. Reinterprets CVOGL as a recurrent localization task, offering a clear conceptual and methodological advance.

2. Employs learnable tokens that iteratively attend to reference features, improving robustness against feature noise.

3. The paper includes a comprehensive set of ablation studies that thoroughly validate the core components of the model.

**Weaknesses:**

1. The Positional Embedding (PE) is mentioned without any detail. It is unclear whether it is a novel design or adopted from prior work. If the latter, a proper citation and explanation are required.

２. The claim in the introduction that ReCOT achieves SOTA performance is questionable. While the method shows strong results on the test set, a closer look at Table 4 reveals that its performance on the validation set for the Ground→Satellite task is suboptimal. For instance, ReCOT's Acc@0.50 on the validation set (43.66%) is lower than both VaGeo (44.42%) and OCGNet (44.20%). A SOTA claim should be consistently supported across both validation and test sets, and the authors should clarify why their method underperforms on the validation set.

３. Multiple formatting and convention errors:
There are several typos and incorrect citations (e.g., "VAGeo" is written as "VaGeo").
Punctuation is used incorrectly in Equation 5.
The dataset is referred to as "CVOGL-DetGeo dataset" in the results section, which is an unconventional and confusing conflation of a dataset name (CVOGL) and a method name (DetGeo).

４. The symbol "n" is used ambiguously throughout the text to denote different concepts (e.g., number of tokens, number of samples).
Variables such as hr, wr are introduced without providing their concrete values in the experiments.
The reporting of model parameters and FPS in Table 4 lacks necessary context. The authors should specify which dataset and task (i.e., Ground→Satellite or Drone→Satellite) these metrics were measured on. Since the two tasks likely involve different input data resolutions, the FPS results could vary significantly between them. Without this clarification, the comparison of efficiency is ambiguous.

5. The core idea of iterative refinement is not new and has been successfully applied to tasks like optical flow (e.g., RAFT) , homography estimation , and feature matching , as the paper itself acknowledges.

6. The use of learnable tokens as queries for iterative refinement is a standard pattern in Transformer-based architectures, and the paper does not sufficiently differentiate its mechanism from existing works in other iterative tasks.

7. Evaluation is conducted only on the CVOGL-DetGeo dataset; broader benchmarks need to be tested. Some inconsistencies and unfair comparisons in experimental setting seriously undermine experimental results' credibility.

8. Some of the references are incomplete. For example, the references on Lines 528-533 lacks essential source information, such as the publication venue and page numbers.

**Questions:**

１.Severe inconsistencies in experimental parameters. The description of the core hyperparameter, the number of recurrent steps m, is contradictory across the paper.
In the ablation study (Sec. 6, discussion of Table 1), the authors state they selected m=5 for the Ground→Satellite task, while m = 3 for the Drone→Satellite task.
Confusingly, in the discussion of qualitative results (Sec. 7), this changes again to we set the number of recurrent steps to 5 for Ground→Satellite and 4 for Drone→Satellite.
Such conflicting statements about a central parameter make it impossible for reviewers to determine the actual settings used in the main comparison experiments (Table 4), rendering the reported results are very confusing..

２. The comparison with baseline methods appears to be unfair.
The Implementation Details mention that the proposed method was trained for 300 epochs.　In contrast, the original papers for the baseline methods (e.g., DetGeo, VaGeo, OCGNet) report training for a much shorter duration, typically 24 epochs.

３. In Section 3.1, the authors state the view that "recurrent strategies have shown superior robustness across domains". Inspired by the aforementioned view, the paper proposes ReCOT. However, the ablation study on different steps m in Table 1 does not seem to prove the effectiveness of the proposed method. For instance, on the Drone→Satellite task, as the number of steps increases, the experimental performance shows almost no improvement, and even degrades on the Acc@0.25 metric.
Although the authors attribute this phenomenon to "overshoot," this contradicts the paper's motivation of "iterative refinement strategies." Especially on the Drone→Satellite task, m=1 appears to offer better overall performance. This significantly diminishes the practical appeal and the claimed novelty of the proposed recurrent framework.

４. The ablation study on the auxiliary losses (LSAM and LToken) is incomplete and presents a logical contradiction.
Table 5 shows that setting the balancing coefficient α=0.1 results in almost no performance drop on the Ground→Satellite task. This suggests that the auxiliary losses have a very small impact.
However, Table 2 shows that removing either LSAM or LToken individually leads to a significant performance degradation.
This creates a paradox: why do these components have a strong effect when removed one by one, but their combined weight can be reduced to almost zero without penalty? To resolve this, the authors must provide an experiment with α=0 (completely removing both auxiliary losses) and analyze the non-linear interaction that explains this phenomenon.

5. The design choice for injecting prompt information is unclear and insufficiently justified. Why is the Prompt Embedding (P) fused with the query features after the Image Encoder, specifically through cross-attention with Fqc?
Existing SOTA works like DetGeo and VaGeo typically inject positional information before the Image Encoder to guide feature extraction from the outset. Alternatively, the prompt information could have been injected at the MHSA stage to directly provide positional guidance to the query tokens (Tq).
The paper does not provide an ablation study or a clear rationale for why its chosen post-encoder fusion strategy is superior to these more conventional and seemingly more direct approaches.

6. It is unclear why this complex, indirect, segmentation-style loss is superior to or necessary in addition to the main DETR-style localization loss $\mathcal{L}_{local}$, which already provides supervision at each recurrent step.

7. The introduction of the auxiliary $\mathcal{L}_{Token}$ loss adds complexity without a clear and compelling justification.

---

### Note · Authors · 2025-11-12

I have read and agree with the venue's withdrawal policy on behalf of myself and my co-authors.